# Phenotypic Variability of Root System Architecture Traits for Drought Tolerance among Accessions of Citron Watermelon (*Citrullus lanatus* var. *citroides* (L.H. Bailey)

**DOI:** 10.3390/plants11192522

**Published:** 2022-09-26

**Authors:** Takudzwa Mandizvo, Alfred Oduor Odindo, Jacob Mashilo, Julia Sibiya, Sascha Lynn Beck-Pay

**Affiliations:** 1School of Agricultural, Earth and Environmental Sciences, University of KwaZulu-Natal, Private BagX01, Scottsville, Pietermaritzburg 3209, South Africa; 2Limpopo Department of Agriculture and Rural Development, Agriculture Regulatory and Technology Development, Directorate, Towoomba Research Centre, Private Bag X1615, Bela-Bela 0480, South Africa

**Keywords:** biomass partitioning, digital root phenotyping, image analysis, rhizotron, root architecture, root phenes, RootSnap

## Abstract

Citron watermelon (*Citrullus lanatus* var. *citroides*) is a drought-tolerant cucurbit crop widely grown in sub-Saharan Africa in arid and semi-arid environments. The species is a C3 xerophyte used for multiple purposes, including intercropping with maize, and has a deep taproot system. The deep taproot system plays a key role in the species’ adaptation to dry conditions. Understanding the root system development of this crop could be useful to identify traits for breeding water-use efficient and drought-tolerant varieties. This study compared the root system architecture of citron watermelon accessions under water-stress conditions. Nine selected and drought-tolerant citron watermelon accessions were grown under non-stress (NS) and water stress (WS) conditions using the root rhizotron procedure in a glasshouse. The following root system architecture (RSA) traits were measured: root system width (RSW), root system depth (RSD), convex hull area (CHA), total root length (TRL), root branch count (RBC), total root volume (TRV), leaf area (LA), leaf number (LN), first seminal root length (FSRL), seminal root angle (SRA), root dry mass (RDM), shoot dry mass (SDM), root–shoot mass ratio (RSM), root mass ratio (RMR), shoot mass ratio (SMR) and root tissue density (RTD). The data collected on RSA traits were subjected to an analysis of variance (ANOVA), correlation and principal component analyses. ANOVA revealed a significant (*p* < 0.05) accession × water stress interaction effect for studied RSA traits. Under WS, RDM exhibited significant and positive correlations with RSM (r = 0.65), RMR (r = 0.66), RSD (r = 0.66), TRL (r = 0.60), RBC (r = 0.72), FSRL (r = 0.73) and LN (r = 0.70). The principal component analysis revealed high loading scores for the following RSA traits: RSW (0.89), RSD (0.97), TRL (0.99), TRV (0.90), TRL (0.99), RMR (0.96) and RDM (0.76). In conclusion, the study has shown that the identified RSA traits could be useful in crop improvement programmes for citron watermelon genotypes with enhanced drought adaptation for improved yield performance under drought-prone environments.

## 1. Introduction

Citron watermelon (*Citrullus lanatus* var. *citroides* (L.H. Bailey)) is an important cucurbit crop grown for multiple purposes such as human food and animal feed in many parts of Africa. Several plant parts of the crop are consumed for food, namely: fresh leaves, ripened fruit and seed, which provide essential nutrients and phytochemical compounds [1,2,3]. Fresh or dried vines are used as feed for domesticated animals [3,4]. Citron watermelon is the most drought and heat tolerant of the cucurbit crops [4]. The crop is also efficient at absorbing N [5,6]; and is tolerant to biotic stresses, including pathogenic diseases such as *Fusarium* wilt, gummy stem blight, bacterial fruit blotch, powdery mildew, viral diseases and root-knot nematodes [7,8,9]. Due to these desirable attributes, citron watermelon is presently being explored as a suitable rootstock for improving biotic and abiotic stress tolerance and fruit quality of grafted sweet watermelon [10,11]. In addition, citron watermelon is a preferred cucurbit crop for gene introgression and breeding in sweet watermelon.

Citron watermelon has a deeper and more well-developed root system (high root branch count, root length and convex hull area) than most cucurbit crops, including sweet watermelon (*Citrullus lanatus* subsp. *vulgaris* Achigan-Dako 06NIA 224 (GAT) Benin), tendril-less melon (*Citrullus ecirrhosus* Griffin 16056 (M) Namibia) and Egusi watermelon (*Citrullus mucosospermus* Vavilov CIT 204 (GAT) Benin) [10,12]. This well-developed root system may explain the species’ ability to tolerate drought conditions and produce an optimum fruit yield in drought-prone environments. Therefore, understanding root morphology development in this species in water-restricted environments will add useful information for improved yield performance. The root system architecture (RSA) has a high degree of plasticity, allowing the plant to acclimate to changing environmental conditions [13,14]. Plant plasticity is important to competitiveness and resilience to changing environmental conditions [14,15]. Soil moisture is an important environmental factor that impacts RSA traits. For example, sweet watermelon has been reported to have enhanced root development under low soil moisture conditions [16]. In citron watermelon, little information is available regarding RSA development and how soil moisture impacts the development of below-ground plant organs.

The citron watermelon root architecture has a primary taproot and several lateral roots [6]. Genetic variability has been reported in the species for morphological traits, including RSA traits [17,18,19,20,21,22,23,24,25]. Using a genetically diverse population of citron watermelon accessions collected and grown in the drier parts of South Africa by small-holder farmers, Mandizvo et al. [26] observed variability in the estimated root percentage, taproot length and root dry mass among the accessions after exposure to drought stress. The authors observed that some accessions maintained either a lower or higher root biomass independent of growth conditions (e.g., non-stressed or water-stressed), suggesting a substantial genetic control of RSA traits in citron watermelon.

The observed phenotypic variability in citron watermelon RSA traits suggests that there could be underlying genetic variation among citron watermelon landraces in relation to root morphology development and architecture under drought conditions. Therefore, understanding root system development under drought conditions in this species could aid in breeding high-yielding, and improved cultivars with enhanced water-use and drought-tolerance traits adapted to dry conditions, which are further exacerbated by prolonged dry spells and erratic rainfall as a result of climate change. The objective of this work was to study the root system architecture of citron watermelon accessions and identify drought-adaptive root traits for cultivar improvement under water-stressed environments.

## 2. Results

### 2.1. Gravimetric Water

In Figure 1, the rate of exponential moisture loss was higher in sand soil (0.131% day^−1^) compared to a mix of sand-pine bark mix (0.094% day^−1^). It took 9 days for the sandy soil to lose 60% of the soil water content, while it took 14 days for the sand–pine bark mix to lose the same amount (60%) of soil moisture. At 18 days after saturation, the sand soil had reached the permanent wilting point (PWP). It took ten days more for the sand–pine bark mix to reach PWP status (Figure 1). Mixing sand with pine bark (Gromor potting Mix 30 dm^3^) improved the water-holding capacity of the sand. Therefore, based on the soil-moisture curve(s) in Figure 1, the Gromor potting Mix 30 dm^3^ and filter sand, mixed in a ratio of 1:3, were used as a growth substrate.

### 2.2. Root Growth of Citron Watermelon Accessions under Non-Stress and Water Stress Conditions

Under NS conditions, the average growth rate of RSW was highest in WWM-76 (0.386 cm day^−1^) and lowest in WWM-15 (0.191 cm day^−1^) (Figure 2a). In Figure 2c, higher average growth rates of RSD (≥1.045 cm day^−1^) were recorded in WWM-76, WWM-41(A) and WWM-39, while lower rates (≤0.845 cm day^−1^) were recorded in WWM-68, WWM-15 and WWM-37(2). Convex hull area of the root system increased at a higher rate (≥60.933 cm^2^ day^−1^) in WWM-76, WWM-09 and WWM-41(A) as compared with WWM-64, WWM-46 and WWM-68 (≤31.715 cm^2^ day^−1^) (Figure 2e). The average growth rate of TRL was higher in WWM-39, WWM-37(2), and WWM-41(A) (≥2.207 cm day^−1^) and lower in WWM-15, WWM-46, and WWM-68 (≤1.670 cm day^−1^) (Figure 2g). Accessions WWM-09, WWM-41(A) and WWM-76 had average RBC ≈ 2 branches day^−1^, while WWM-15, WWM-37(2), WWM-39 and WWM-68 were forming approximately one branch per day (Figure 2i). Accessions WWM-09, WWM-37(2) and WWM-41(A) had higher leaf area expansion rates ≥1.987 cm^2^ day^−1^. Lower leaf expansion rates (≤1.731 cm^2^ day^−1^) were recorded in WWM-39, WWM-46 and WWM-68 (Figure 2k; Table 1).

Under WS conditions, the average growth rate of RSW was higher in WWM-76, WWM-41(A) and WWM-37(2) (≥0.325 cm day^−1^) and lower in WWM-64, WWM-15 and WWM-68 (≤0.284 cm day^−1^) (Figure 2b). In Figure 2d, higher average growth rates of RSD (≥1.152 cm day^−1^) were recorded in WWM-09, WWM-41(A) and WWM-76, while lower rates (≤0.889 cm day^−1^) were recorded in WWM-15, WWM-46 and WWM-68. Convex hull area of the root system increased at a higher rate (≥78.593 cm^2^ day^−1^) in WWM-76, WWM-41(A) and WWM-09 compared to WWM-15, WWM-46, WWM-64 and WWM-68 (≤41.477 cm^2^ day^−1^) (Figure 2f). The average growth rate of TRL was higher in WWM-09, WWM-37(2), WWM-41(A) and WWM-76 (≥2.207 cm day^−1^) and lower in WWM-15, WWM-46, WWM-68 and WWM-64 (≤1.779 cm day^−1^) (Figure 2h). WWM-41(A), WWM-68 and WWM-76 had higher leaf area expansion rates ≥0.804 cm^2^ day^−1^. Lower leaf expansion rates (≤0.403 cm^2^ day^−1^) were recorded in WWM-15, WWM-46 and WWM-64 (Figure 2l). From 21 DAP, the leaf-area growth curves under WS conditions started to plateau and decline in accessions such as WWM-15, WWM-46 and WWM-68 due to water deficit (Figure 2l; Table 1).

Water stress increased the mean growth rate of RSW (0.298 cm day^−1^) as compared with NS conditions (0.273 cm day^−1^) (Figure 3a). Both RSD (Figure 3b) and CHA (Figure 3c) average growth rates for all evaluated accessions were higher under WS condition (1.102 cm day^−1^ and 60.276 cm^2^ day^−1^) than under NS conditions (0.909 cm day^−1^ and 48.044 cm^2^ day^−1^) respectively. Mean total root length growth rate did not differ significantly among the accessions under both the NS and WS conditions; the rates of TRL growth ranged between 1.670–2.207 cm day^−1^ under NS and between 1.779–2.325 cm day^−1^ under WS conditions (Figure 3d). The average rate of root branch count (≈1 branch day^−1^) did not differ between water treatments (Figure 3e). In Figure 3f, the average leaf-area expansion rate was higher under NS conditions (1.909 cm^2^ day^−1^) than under WS conditions (0.762 cm^2^ day^−1^).

### 2.3. ANOVA Showing Accession, Water Regime and Their Interactions on Root and Shoot Traits of Citron Watermelon

ANOVA for evaluated root traits indicated that the effects of irrigation, genotype, and interaction were significantly different for most traits (Table 2). Water stress significantly increased the average RSW, RSD, CHA, TRV, RSM and RMR as compared with non-stress conditions (Figure 4a–c,f,m,n). Accessions WWM-09, WWM-41(A) and WWM-76 recorded RSW values of ≥10.940 cm under WS conditions (Table 3), whereas accessions WWM-68, WWM-39, WWM-15 and WWM-46 recorded RSW values of ≤8.644 cm under NS conditions (Table 4).

For RSD, accessions WWM-09, WWM-41(A) and WWM-76 recorded values ≥ 34.820 cm under WS, compared to WWM-15, WWM-46, WWM-68 and WWM-64, which recorded RSD values ≤ 28.770 cm under NS conditions. The mean CHA for evaluated accessions was significantly higher (1620.111 cm^2^) under WS conditions compared with NS conditions (1362.566 cm^2^) (Figure 4c). For TRV, WWM-09, WWM-39, WWM-41(A), WWM-37(2), WWM-76 and WWM-68 recorded values ≥ 1.928 cm^3^ under WS, compared to WWM-15 and WWM-46 which recorded TRV values ≤ 1.541 cm^3^ under NS conditions.

The mean root–shoot mass ratio for evaluated genotypes was significantly higher (1.8881) under WS conditions compared to NS conditions (1.2343) (Figure 4m). Mean RMR for evaluated accessions was significantly higher (0.6463 g g^−1^) under WS conditions compared to NS conditions (0.5463 g g^−1^) (Figure 4n).

Water stress significantly reduced mean RBC, LA, LN, RDM, SDM, SMR and RTD among the evaluated accessions compared to NS conditions (Figure 4e,i–l,o,p). The mean RBC for evaluated accessions was higher (43 branches) under NS conditions compared to WS conditions (39 branches) (Figure 4e). The average leaf number for evaluated accessions was significantly higher (8 leaves) under NS conditions compared to WS conditions (5 leaves) (Figure 4j).

For RDM, accessions WWM-41(A), WWM-76, WWM-39 and WWM-09 recorded values ≥ 2.695 g under NS conditions, compared to WWM-15, WWM-37(2), WWM-46 and WWM-68, which recorded RDM ≤ 1.355 g under WS conditions. WWM-09, WWM-39, WWM-64, WWM-41(A), WWM-46 and WWM-76 recorded SDM values of ≥2.172 g under NS conditions, whereas accessions WWM-15, WWM-37(2) and WWM-76 recorded SDM values ≤ 0.747 g under WS conditions. Under NS conditions, a higher SMR (≥0.514 g g^−1^) was recorded in WWM-15 and WWM-64, whereas accessions WWM-09 and WWM-76 recorded a lower SMR (≤0.392 g g^−1^).

Under WS conditions, accessions WWM-15, WWM-64 and WWM-68 recorded a higher SMR (≥0.387 g g^−1^) and accessions WWM-09 and WWM-76 recorded lower values (≤0.285 g g^−1^). The mean root tissue density (RTD) for evaluated accessions was significantly higher (1.7252 g cm^3^) under NS conditions compared with WS conditions (0.8043 g cm^3^) (Figure 4p).

### 2.4. Pearson Correlation Analysis Showing Associations of RSA Traits among Citron Watermelon Accessions under Non-Stressed and Water-Stressed Conditions

Pearson correlation coefficients showing evaluated traits relationships among citron watermelon accessions under non-stress and water stress conditions are presented in Table 5. Under NS conditions, significant and positive correlations were observed between RDM and RBC (r = 0.91; *p* = 0.002), SDM (r = 0.91; *p* = 0.001), RSM (r = 0.92; *p* < 0.001), RMR (r = 0.88; *p* = 0.021) and RTD (r = 0.89; *p* = 0.019). Root branch count was positively correlated with SDM (r = 0.92; *p* < 0.001). Significant and negative correlations were observed between SMR with RTD (r = −0.86; *p* = 0.017), RSM (r = −0.99; *p* < 0.001), SDM (r = −0.63; *p* = 0.048), RDM (r = −0.88; *p* = 0.016), RBC (r = −0.73; *p* = 0.031) and CHA (r = −0.72; *p* = 0.045) (Table 5; *bottom diagonal*). Under water stress conditions, significant and positive correlations were observed between RSW with RSD (r = 0.83; *p* < 0.001), CHA (r = 0.98; *p* < 0.001), TRL (r = 0.90; *p* < 0.01), TRV (r = 0.85; *p* = 0.021), RSM (r = 0.82; *p* = 0.033) and RMR (r = 0.83; *p* = 0.001). Significant and negative correlations were observed between SMR with RSW (r = −0.82; *p* = 0.027), RSD (r = −0.94; *p* = 0.001), CHA (r = −0.91; *p* < 0.001), RDM (r = −0.65; *p* = 0.042), RMR (r = −1.000; *p* < 0.001), TRL (r = −0.93; *p* = 0.009), RBC (r = −0.93; *p* = 0.008), TRV (r = −0.78; *p* = 0.032), FSRL (r = −0.90; *p* < 0.001), LA (r = −0.73; *p* = 0.027) and RSM (r = −0.99; *p* < 0.001) (Table 5; *top diagonal*).

### 2.5. Principal Component Analysis (PCA) for Root System Architecture of Citron Watermelon Accessions Evaluated under Non-Stressed and Water-Stressed Conditions

Table 6 shows PCA with factor loadings, eigenvalues, and percent variance of the evaluated RSA traits of nine selected drought-tolerant accessions under non-stressed and water-stressed conditions. Under NS conditions, PC1 accounted for 63.95% of the total variation and was positively correlated with RSW, RSD, TRV, RDM CHA, RMR and RTD. PC2 positively correlated with TRL, LA and SRA, contributing 14.81% of the total variation. Under WS conditions, PC1 accounted for 64.50% of the total variation and was positively correlated with RSD, RSW, CHA, TRL, RBC, TRV, FSRL, RMR, RSM, LA, RSM and RMR. Leaf numbers RDM, SDM and RTD were positively correlated with PC2, which accounted for 22.85% of the total variation (Table 6).

The PC biplots based on PCA analysis were used to visualize the relationship between citron watermelon accessions and root and leaf traits under NS (Figure 5a) and WS conditions (Figure 5b). Traits represented by parallel vectors or close to each other revealed a strong positive association, and those located nearly opposite (at 180°) showed a highly negative association, while the vectors toward sides expressed a weak relationship. Under NS conditions, accessions WWM-09, WWM-39 and WWM-76 are grouped based on high RBC, SDM, RSM, LN and RTD. Accessions WWM-37(2) and WWM-15 are grouped based on high SMR (Figure 5a). Under WS conditions, accessions WWM-09, WWM-39 and WWM-41(A) are grouped based on high RBC, RDM, RSM, RSD, RMR and FSRL. Accessions WWM-37(2) and WWM-76 are grouped based on high TRL, LA, CHA, RSW and TRV. WWM-46, WWM-68 and WWM-15 are grouped based on high SMR (Figure 5b).

### 2.6. Root Vigour (Foraging Capacity) of Citron Watermelon Accessions under Water Stress Conditions

Agglomerative hierarchical clustering for the means of root traits under water stress conditions at 35 DAP (Table 3) classified the nine landrace accessions into four groups (Figure 6). Group A (high root foraging) comprised one accession (WWM-76). Group B comprised two accessions (WWM-09 and WWM-41(A)) with moderate-high root foraging. Similarly, group C had two accessions (WWM-37(2) and WWM-39) with moderately low root foraging. Group D (low root foraging) was comprised of four accessions (WWM-15, WWM-46, WWM-64 and WWM-68) (Figure 6).

## 3. Discussion

The present study determined the root system architecture of citron watermelon accessions to aid in the selection of key drought-adaptive root traits for breeding targeting water-stressed environments. Root system architecture plays an important role in citron watermelon’s response to water stress [10]. The roots are the first plant organs to respond to water stress. In the present study, the variation in RSA traits among citron watermelon accessions under non-stressed (Table 4) and water-stressed conditions (Table 3) indicates substantial genetic variability for efficient selection of root-adaptive traits to drought stress. Some important RSA traits for enhanced water-uptake include root system length and width, convex hull area, root branch count and total root length [10]. In the present study, citron watermelon accessions such as WWM-37(2), WWM-41(A) and WWM-76 increased total root length, convex hull area, root system width and total root volume indicating their ability to absorb water under water stress conditions. This agreed with Katuuramu, Wechter, Washington, Horry, Cutulle, Jarret and Levi’s [10] results that total root length, average root diameter, total root surface area and total root volume are important RSA traits for adaptation to drought stress in *C. lanatus*, including sweet and citron watermelons.

On the contrary, according to our results, not all accessions evaluated in the present study had increased root length under water stress (Table 3). This contradicts the widely generalised view that total root length increases in drier environments [27,28,29,30]. On the contrary, Schenk and Jackson [31] highlighted that water availability is not the only abiotic influencing rooting depth. Soil texture and genotype composition will also dictate the total root length. The root system architecture is a function of both genetic endogenous programmes (regulating growth and organogenesis) and the action of edaphic environmental stimuli. This is supported by a significant interaction between accessions and water conditions.

The efficient response of the root system of the evaluated accessions is also supported by their higher leaf number and shoot biomass compared to other tested accessions (Table 6). The present study agrees with Guzzon et al. [32] that citron watermelon exhibit higher above-and-below ground biomass under water deficit conditions as a drought-avoidance strategy. Therefore, the identified RSA traits are recommended for selecting and highly breeding drought-tolerant citron watermelon cultivars in the stir of increased weather conditions in the future. Also, the present findings suggest that citron watermelon can be a donor of root traits for introgression in close related cucurbit species including sweet dessert watermelon to improve drought tolerance and adaption in water-limited environments.

The shift in root growth and allometry observed in the present study can be explained by the “balanced growth” hypothesis, which states that plants respond to drought by promoting or maintaining root growth while reducing shoot growth [33,34,35]. Mandizvo and Odindo [36] reported high partitioning of dry matter to roots than shoots of Bambara groundnuts (*Vigna subterranea* L.) even under environmental stimuli (nutrient deficiency). Increased root versus shoot growth improved citron watermelon hydraulic status under water stress conditions, probably due to (i) increased root to leaf surface, (ii) continued production of new root tips and (iii) enhancement of plant capacity for acquiring water to support the development of existing shoots. A high root-to-shoot ratio is important; a greater root-to-shoot ratio means greater root density and root interception for water uptake [37]. Variations in the root-to-shoot ratio have been previously reported in citron watermelons, whereby drought tolerant citron watermelon genotypes showed higher values [32]. Similar to the present findings, citron watermelon accessions WWM-09, WWM-39 and WWM-41(A) had higher root-to-shoot ratios indicating higher levels of drought tolerance.

Mandizvo, Odindo, Mashilo and Magwaza [26] highlighted that, as the soil water starts depleting, prolific and deep root systems accompanied by the maintenance of leaf surface area is a key attribute of drought-tolerance in citron watermelon. This is supported by positive correlation between root tissue density with shoot biomass (r = 0.84) and leaf number (r = 0.79) (Table 5). These observations agree with present findings, indicating that citron watermelon develops a deep root system to allow deep water access and produce high biomass under water-constrained environments. As evidenced by negative associations formed in PC biplots (Figure 5b) between SMR with TRL, CHA, RSD and RSW; drought stress induced a conservative balance between water-losing organs (leaves) and water-gaining organs (roots) in the evaluated citron watermelon accessions.

Some RSA traits, including the deep root system of citron watermelon, are preferred rootstock for improving fruit and quality of grafted sweet watermelon for dry water-limited environments [6,38,39]. Understanding the interrelationships among below ground (root) growth, above ground (shoot) growth and allometry can provide useful information for an integrated drought tolerance breeding approach. The positive associations observed between root–shoot mass ratio and various root traits under water stress conditions, including root system width, root system depth, total root length, convex hull area and total root volume, suggested synchronised selection and improvement of these traits in citron watermelon.

## 4. Materials and Methods

### 4.1. Plant Material

The Department of Agriculture and Rural Development (DARD), Bela-Bela, Limpopo Province, South Africa, provided citron watermelon accessions for the study. Out of forty citron watermelon accessions, nine accessions classified as “*highly drought-tolerant*” by Agglomerative Hierarchical Clustering (AHC) using the six drought indices from our previous study [26] for root phenotyping. Based on previous study findings by Mandizvo, Odindo, Mashilo and Magwaza [26], each accession’s drought stress tolerance index is summarised in Table 7.

### 4.2. Fabrication Rhizotron Prototype

A root rhizotron was fabricated following the method described by Wiese et al. [40]. Transparent Impex Polycarb sheets of 3 mm thickness, purchased from Maizey Private Limited, Pietermaritzburg, South Africa, were cut into rectangular sheets (R_1_ and R_4_) measuring 50 cm in length and 30 cm in width using a table saw (Ryobi, Hiroshima, Japan). Wooden boards of 12 mm thickness were cut into rectangular planks measuring 50 cm in length and 3 cm in width (R_2_ and R_3_). All the cut rhizotron pieces (R_1_, R_2_, R_3_ and R_4_) were held together using a Grip GV9365 Bench Vice (100 mm) to allow the drilling of aligned pilot holes (Figure 7).

Holes of 8 mm ⌀ were drilled on each rhizotron piece on a flat surface using Ryobi 16 mm bench drill press. Rhizotron pieces were assembled and secured using an adhesive (NO MORE NAILS, Pattex^®^), cable ties (T5OI 4.8 × 300 mm) and brown buff packaging tape. Each lateral side of the rhizotron was used to evaluate different systems for non-disruptive visualisation of roots while holding the substrate in place (Figure 7). On average, each rhizotron weighed ≈ 0.948 ± 0.038 kg, enclosing ≈ 1.8 × 10^−3^ m^3^ of soil. The estimated cost for a single unit of rhizotron was ZAR114.30/USD7.00 (Appendix A).

### 4.3. Growth Substrate Selection

The substrate was selected based on the gravimetric water content (*θ_g_*) of (i) filter sand, (ii) Gromor potting Mix 30 dm^3^ (pine bark) and (iii) a mix of Gromor potting Mix 30 dm^3^ and filter sand mixed in ratio a 1:3. Each of the three substrates was filled in a rhizotron weighing (0.948 ± 0.038 kg). The substrate was transferred into a ceramic bowl and dried in an oven at 105 °C for 24 h. The mass of dry soil was determined by subtracting the mass of empty rhizotrons from the sum mass of oven-dry soil and rhizotron. The substrate in each rhizotron was watered to saturation and left to drain freely through percolation. The change in rhizotron weight was measured daily using a sensitive electron balance (Adam AAA 100L, Adam Equipment, Gauteng, South Africa) for 35 days. The (*θ_g_*) of each substrate was calculated according to Haney and Haney [41] (Equation (1)). Based on these results, a mix of Gromor potting Mix 30 dm^3^ and filter sand mixed in a ratio of 1:3 was used for the present study.
(1)$$\left(\theta g\right)\;\;\left(\%\right)=\left[\frac{Mass\;of\;moist\;soil\;{\left(g\right)}_{\left(saturated\right)}-Mass\;of\;oven\;dried\;soil\;\left(g\right)}{Mass\;of\;oven\;dried\;soil\;\left(g\right)}\right]\times 100$$

### 4.4. Experimental Design and Growth Conditions

Root rhizotron experiments were done under glasshouse conditions at the Controlled Research Facility (CEF) of the University of KwaZulu-Natal, Pietermaritzburg, South Africa (29°37′37.5″ S and 30°24′10.4″ E). The glasshouse’s mean air temperature and relative humidity were 25 ± 2 °C and 60 ± 3%, respectively. The first rhizotron experiment was conducted between September and October 2021, and the second between October and November 2021. The study was designed as a 9 × 2 factorial experiment with 9 citron watermelon accessions grown under two water regimes: non-stressed (NS) and water-stressed (WS). The experiment was laid in a completely randomized design (CRD) with three replications, giving 54 experimental units (1.8 × 10^−3^ m^3^ rhizotrons). One seed of each accession was sowed in a rhizotron filled with a weed-free Gromor potting Mix 30 dm^3^ and filter sand mixed in a ratio of 1:3. Plants under NS were irrigated at planting, 14 days after planting (DAP) and 28 DAP. For WS treatment, irrigation was done at planting only. The soil-moisture curve (Figure 1) was used to estimate soil water content throughout the experiment. Each lateral side of the rhizotron was covered with black polyethylene plastic to simulate darkness and avoid light-induced root growth. Two-dimensional root images were captured from 8 to 35 DAP using the method described in Section 4.5. Leaf images were also captured to monitor changes in leaf area. The experiment was harvested at 35 DAP. The roots and shoots were separated and dried in an oven at 70 °C for 24 h. A precision scale (UW4200H Shimadzu, Kyto, Japan) was used to measure root and shoot dry mass.

### 4.5. Image Acquisition

A camera positioning technique was used to hold the camera at a constant distance (80 cm) from the rhizotron for time-series digital capturing of root growth. Images were captured on both lateral sides of the rhizotron daily from 8 to 35 DAP. An AI camera of 16 megapixels (Huawei Y9 Prime 2019, Huawei Technologies Co., Ltd, Guangdong, China) was used to capture images. Camera settings included a resolution of 4:3, assistive grid on, and a timer of 3 s. Images were collected in the raw format with a colour depth of 12 bits and an image size of 4288 × 2848 pixels. Leaf area was measured using the Easy Leaf Area smartphone application (Heaslon, University of California, California) described by Easlon and Bloom [42].

### 4.6. Image Analysis and Data Collection

A software package (RootSnap Version 1.3.2.25, CID Bio-Science Inc., Camas, WA USA) analysed the 2D images of the plant roots captured from rhizotron laterals. The software performed the predictions in automatic mode with manual corrections (Figure 8). Root system architectural traits (Table 8) from captured root images were quantified using a user-assisted root image analysis package (RootSnap Version 1.3.2.25, CID Bio-Science Inc.) on a computer tablet (Microsoft Surface). A Microsoft Surface Pro 4 stylus was used to trace the roots. Continuous data from 8 to 35 DAP was collected for root system width (RSW), root system depth (RSD), convex hull area (CHA), total root length (TRL), root branch count (RBC) and leaf area (LA). Other root traits summarised in Table 2 were measured after harvesting the experiment (35 DAP).

### 4.7. Statistical Analysis

Analysis of variance was performed for traits measured using Genstat 20th Edition (VSN International, Hempstead, UK). Means were separated using Fisher’s protected least significant difference (LSD) test when treatments showed significant effects on measured parameters at a 5% level of significance. Principal component analysis (PCA) and the biplot diagrams were exploited using Origin Pro 2021b (OriginLab Corporation Northampton, Massachusetts). Pearson correlations were computed based on mean values using GraphPad Prism Version 9.2.0 (GraphPad Software, Inc., San Diego, CA, USA). Agglomerative Hierarchical Clustering (AHC) was done according to Ward’s method using squared Euclidean distance to measure similarity using XLSTAT.

## 5. Conclusions

The present study compared the root system architecture of drought-tolerant citron watermelon accessions to aid the cultivation of efficient drought-adaptive root traits for cultivar development in water-stressed environments. Using water as a limiting edaphic factor, this study showed that plasticity and biomass allocation shift in different ways according to genotype, presumably to optimise the use of limited resources. The study found significant phenotypic variation in root architecture among citron watermelon accessions that may relate to differences in water uptake. The following RSA traits, including total root length, root system width, convex hull area and total root volume, were associated with drought tolerance. Further, RSA traits such as root dry mass and root shoot mass ratio were highly correlated with root branch count, root system depth, total root length and leaf number. These traits are useful selection criteria for breeding and developing water-efficient citron watermelon accessions for cultivation in drought-prone environments.

## Figures and Tables

**Figure 1 plants-11-02522-f001:**
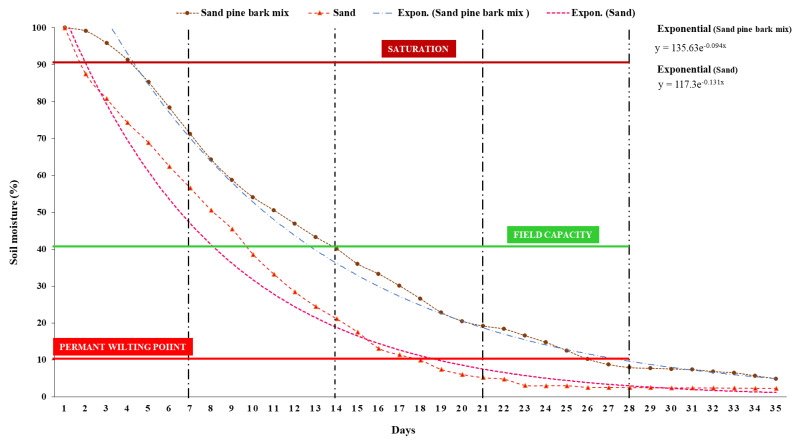
Percentage of soil water content depleted versus time (days) in sand soil, pine bark and sand–pine bark mix.

**Figure 2 plants-11-02522-f002:**
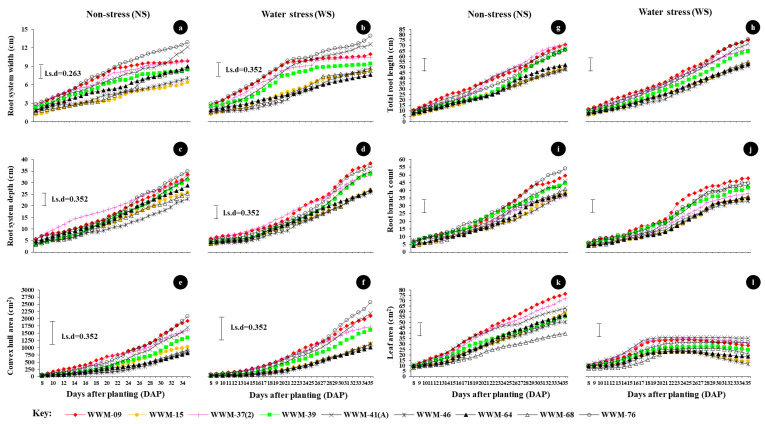
Changes in root growth and leaf area of nine drought-tolerant citron watermelon accessions under non-stressed and water stress conditions from 8 to 35 days after planting.

**Figure 3 plants-11-02522-f003:**
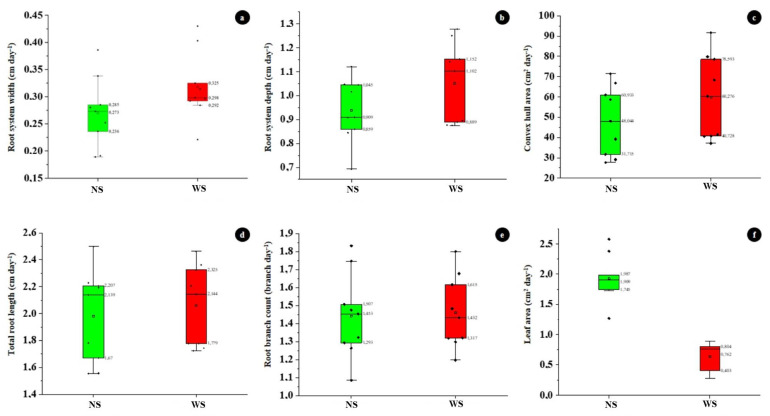
Growth rate comparison of root traits and leaf area under non-stress and water stress conditions (**a**) root system width, (**b**) root system depth, (**c**) convex hull area, (**d**) total root length, (**e**) root branch count and (**f**) leaf area.

**Figure 4 plants-11-02522-f004:**
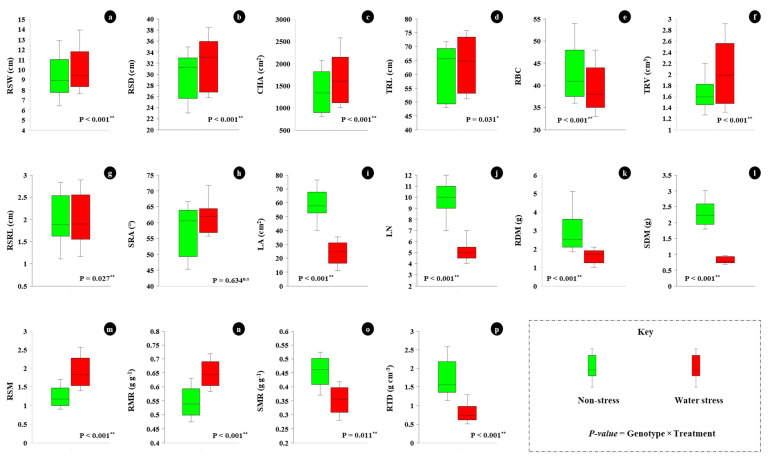
Summarized effect of non-water stress and water stress at 35 DAP: (**a**) root system width, (**b**) root system depth, (**c**) convex hull area, (**d**) total root length, (**e**) root branch count, (**f**) total root volume, (**g**) first seminal root length, (**h**) seminal root angle, (**i**) leaf area, (**j**) leaf number, (**k**) root dry mass, (**l**) shoot dry mass, (**m**) root shoot mass ratio, (**n**) root mass ratio, (**o**) shoot mass ratio, (**p**) root tissue density.

**Figure 5 plants-11-02522-f005:**
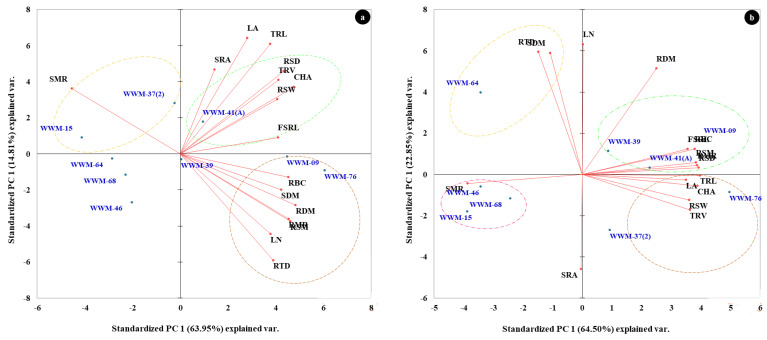
Principal component (PC) biplot of PC 1 vs. PC 2 demonstrating the relationship between root and leaf traits of nine citron watermelon accessions evaluated in rhizotrons under (**a**) non-stress and (**b**) water stress conditions.

**Figure 6 plants-11-02522-f006:**
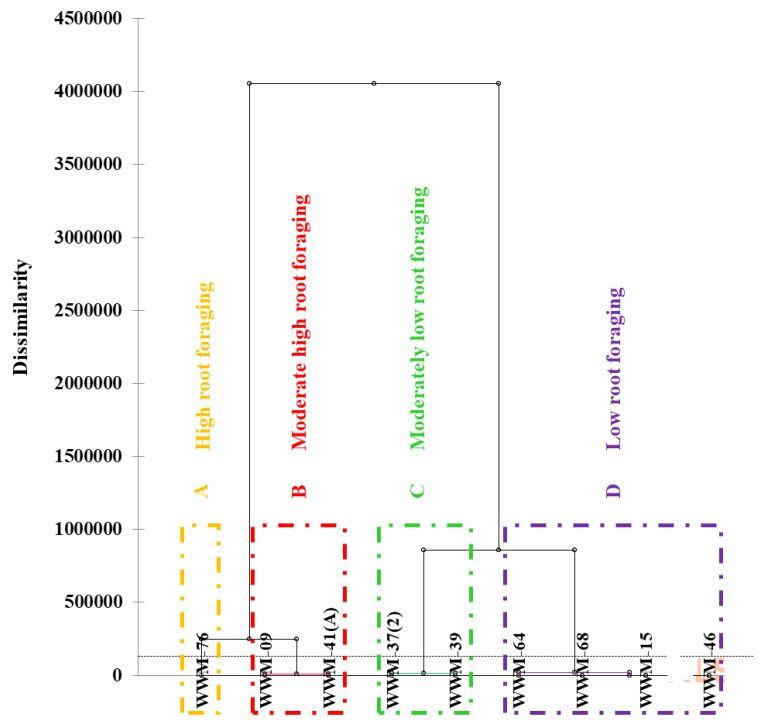
Dendrogram distinguishing the levels of root foraging among nine citron watermelon accessions based on measured root system architecture traits under water stress conditions.

**Figure 7 plants-11-02522-f007:**
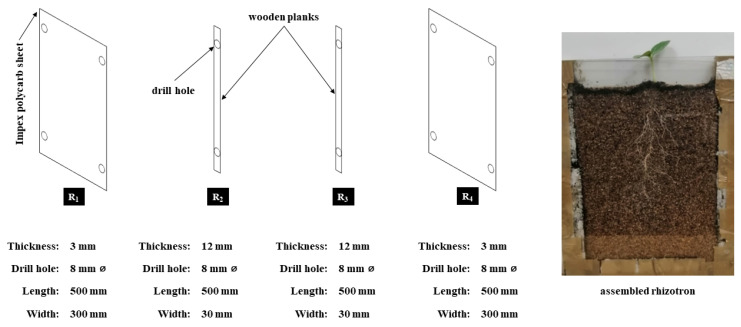
Sketch of an inexpensive rhizotron design assembly—observations of the root systems are taken on the lateral sides of the rhizotrons.

**Figure 8 plants-11-02522-f008:**
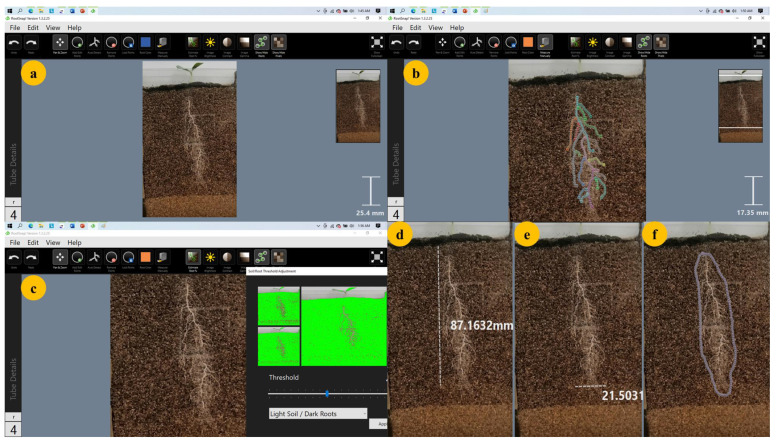
Illustration of how RootSnap software was used to analyse and collect data from captured root images: (**a**) root image in raw format imported from local storage to RootSnap; (**b**) tracing the root using Microsoft Surface Pro 4 stylus to measure total root length; (**c**) automated digital image analysis mode; (**d**) measurement of root system depth; (**e**) measurement of root system width; (**f**) measurement of root convex hull area.

**Table 1 plants-11-02522-t001:** Average daily growth rates of root traits and leaf area measured in nine citron watermelon accessions from 8 to 35 DAP under non-stress and water-stress conditions.

	RSW (cm day^−1^)		RSD (cm day^−1^)		CHA (cm^2^ day^−1^)		TRL (cm day^−1^)		RBC (branch/day)		LA (cm^2^ day^−1^)	
Accession	NS	WS	NS	WS	NS	WS	NS	WS	NS	WS	NS	WS
WWM-09	0.285 ^c^	0.314 ^d^	1.016 ^b^	1.278 ^a^	66.743 ^ab^	78.593 ^b^	2.192 ^bc^	2.325 ^b^	1.746 ^f^	1.800 ^e^	2.578 ^a^	0.762 ^b^
WWM-15	0.191 ^h^	0.284 ^g^	0.859 ^c^	0.889 ^d^	39.172 ^e^	40.728 ^e^	1.670 ^e^	1.779 ^d^	1.263 ^b^	1.298 ^b^	1.909 ^d^	0.278 ^e^
WWM-37(2)	0.273 ^e^	0.325 ^c^	0.909 ^c^	1.102 ^c^	58.729 ^c^	68.341 ^c^	2.499 ^a^	2.207 ^c^	1.293 ^bc^	1.432 ^c^	2.379 ^b^	0.788 ^b^
WWM-39	0.236 ^g^	0.298 ^e^	1.047 ^ab^	1.141 ^bc^	48.044 ^d^	60.276 ^d^	2.207 ^bc^	2.144 ^c^	1.475 ^de^	1.482 ^c^	1.731 ^e^	0.667 ^c^
WWM-41(A)	0.338 ^b^	0.430 ^a^	1.045 ^ab^	1.152 ^bc^	60.933 ^bc^	79.814 ^b^	2.228 ^b^	2.362 ^ab^	1.507 ^e^	1.677 ^d^	1.987 ^c^	0.896 ^a^
WWM-46	0.189 ^h^	0.296 ^e^	0.693 ^d^	0.875 ^d^	29.129 ^f^	40.474 ^e^	1.557 ^f^	1.724 ^d^	1.085 ^a^	1.197 ^a^	1.745 ^e^	0.277 ^e^
WWM-64	0.252 ^f^	0.221 ^h^	0.908 ^c^	0.896 ^d^	27.663 ^f^	37.138 ^e^	1.783 ^d^	1.745 ^d^	1.322 ^c^	1.317 ^b^	1.950 ^cd^	0.403 ^d^
WWM-68	0.280 ^d^	0.292 ^f^	0.845 ^c^	0.877 ^d^	31.715 ^f^	41.477 ^e^	1.553 ^f^	1.779 ^d^	1.453 ^d^	1.319 ^b^	1.268 ^f^	0.804 ^b^
WWM-76	0.386 ^a^	0.403 ^b^	1.120 ^a^	1.250 ^ab^	71.403 ^a^	91.609 ^a^	2.139 ^c^	2.463 ^a^	1.832 ^g^	1.615 ^d^	1.783 ^e^	0.894 ^a^
**Mean**	0.270	0.318	0.938	1.051	48.170	59.828	1.981	2.057	1.442	1.460	1.926	0.641
**LSD**	3.431 × 10^−3^	0.003	0.086	0.120	6.124	7.376	0.086	0.100	0.051	0.069	0.067	0.051
**CV (%)**	0.700	0.600	5.300	6.700	7.400	7.200	2.500	2.900	2.100	2.700	2.100	4.700
** *p* ** **-value**	<0.001	<0.001	<0.001	<0.001	<0.001	<0.001	<0.001	<0.001	<0.001	<0.001	<0.001	<0.001

Values in the same column followed by the same letter are not significantly different, while values with different superscript letters are significantly different according to Fisher’s test. **RSW**; root system width, **RSD**; root system depth, **CHA**; convex hull area, **TRL**; total root length, **RBC**; root branch count, **LA**; leaf area. Average growth rate is the coefficient of x in the linear equation (y = mx + c) derived from linear graphs in Figure 2.

**Table 2 plants-11-02522-t002:** Analysis of variance showing mean squares and significant tests for root and leaf of nine citron watermelon landrace accessions evaluated under non-stressed and water-stressed conditions.

Source of Variation	d.f	RSW	RSD	CHA	TRL	RBC	TRV	LA	LN
Accession (A)	8	25.763 **	117.558 **	1.640 × 10^6^ **	578.406 **	183.560 **	1.060 **	4.383 × 10^2^ **	6.292
Water Condition (WC)	1	6.001 **	48.964 **	8.952 × 10^5^ **	106.145 **	150 **	2.212 **	1.665 × 10^4^ **	280.167 *
A × WC	8	1.545 **	5.107 **	3.152 × 10^4^ **	29.877 **	12.625 **	0.163 **	1.255 × 10^2^ **	3.792 *
Residual	36	0.025	0.051	131.900	0.018	0.093	0.023	4.138 × 10^−2^	1.847
**Source of Variation**	**d.f**	**FSRL**	**SRA**	**RDM**	**SDM**	**RSM**	**RMR**	**SMR**	**RTD**
Accession (A)	8	1.892 **	141.800 ^ns^	2.875 **	0.296 **	0.693 **	0.015 **	0.015 **	0.520 **
Water Condition (WC)	1	0.056	197.100 ^ns^	22.970 **	27.549 **	5.771 **	0.135 **	0.135 **	11.447 **
A × WC	8	0.048 **	130.400 *	1.443 **	0.330 **	0.073 *	8.489 × 10^−4^	8.489 × 10^−4^ **	0.388 **
Residual	36	0.007	379.700	0.056	0.016	0.027	5.185 × 10^−4^	5.185 × 10^−4^	0.045

**d.f**; degrees of freedom, **RSW**; root system width, **RSD**; root system depth, **TRL**; total root volume, **RBC**; root branch count, **TRV**; total root volume, **LA**; leaf area, **LN**; leaf number, **FSRL**; first seminal root length, **SRA**; seminal root angle, **RDM**; root dry mass, **SDM**; shoot dry mass, **RSM**; root shoot mass ratio, **RMR**; root mass ratio, **SMR**; shoot mass ratio, **RTD**; root tissue density, **n**; non-significant. ***** and ****** denote significance at 5% and 1% probability levels, respectively.

**Table 3 plants-11-02522-t003:** Mean values for root and leaf traits of nine citron watermelon accessions evaluated in rhizotrons under water-stress conditions 35 days after planting.

	Below Ground								Above Ground		Allometry					
Accession	RSW	RSD	CHA	TRL	RBC	TRV	FSRL	SRA	LA	LN	RDM	SDM	RSM	RMR	SMR	RTD
WWM-09	10.940 ^c^	38.410 ^a^	2102 ^c^	75.230 ^a^	48 ^a^	2.423 ^b^	2.618 ^b^	58.120 ^a^	28.600 ^d^	6.000 ^ab^	2.097 ^a^	0.822 ^cd^	2.558 ^a^	0.718 ^a^	0.281 ^e^	0.866 ^bc^
WWM-15	8.350 ^g^	27.050 ^f^	1129 ^f^	53.250 ^g^	35 ^g^	1.450 ^d^	1.163 ^f^	58.470 ^a^	14.170 ^h^	4.000 ^c^	0.989 ^e^	0.711 ^de^	1.393 ^d^	0.582 ^e^	0.418 ^a^	0.683 ^cde^
WWM-37(2)	10.400 ^d^	33.030 ^e^	1718 ^d^	66.510 ^d^	38 ^e^	2.697 ^ab^	2.285 ^c^	71.700 ^a^	30.740 ^c^	4.000 ^c^	1.355 ^d^	0.747 ^de^	1.827 ^bc^	0.645 ^bcd^	0.355 ^bcd^	0.504 ^e^
WWM-39	9.450 ^e^	34.010 ^d^	1606 ^e^	64.720 ^e^	42 ^d^	1.983 ^c^	2.483 ^b^	61.900 ^a^	24.690 ^e^	5.000 ^bc^	1.916 ^b^	0.960 ^b^	2.005 ^b^	0.666 ^b^	0.334 ^d^	0.975 ^b^
WWM-41(A)	12.600 ^b^	34.820 ^c^	2193 ^b^	71.730 ^c^	43 ^c^	2.401 ^b^	1.899 ^d^	55.670 ^a^	35.430 ^a^	5.000 ^bc^	1.747 ^c^	0.900 ^bc^	1.976 ^b^	0.663 ^bc^	0.338 ^cd^	0.733 ^cd^
WWM-46	8.720 ^f^	25.820 ^h^	1126 ^f^	51.180 ^h^	33 ^h^	1.315 ^d^	1.531 ^e^	65.670 ^a^	11.360 ^i^	5.000 ^bc^	1.258 ^d^	0.771 ^cde^	1.648 ^bcd^	0.620 ^cde^	0.380 ^abc^	0.969 ^b^
WWM-64	7.620 ^h^	26.620 ^g^	1014 ^g^	53.030 ^g^	35 ^g^	1.493 ^d^	1.849 ^d^	55.670 ^a^	18.600 ^g^	7.000 ^a^	1.888 ^b^	1.293 ^a^	1.466 ^cd^	0.594 ^e^	0.407 ^a^	1.279 ^a^
WWM-68	8.270 ^g^	26.950 ^f^	1114 ^f^	55.070 ^f^	36 ^f^	1.928 ^c^	1.570 ^e^	63.030 ^a^	21.070 ^f^	5.000 ^bc^	1.233 ^d^	0.780 ^cde^	1.585 ^cd^	0.613 ^de^	0.387 ^ab^	0.641 ^de^
WWM-76	13.940 ^a^	36.990 ^b^	2579 ^a^	75.800 ^a^	45 ^b^	2.912 ^a^	2.888 ^a^	61.900 ^a^	31.730 ^b^	5.000 ^bc^	1.709 ^c^	0.681 ^e^	2.535 ^a^	0.716 ^a^	0.285 ^e^	0.589 ^de^
l.s.d	0.120	0.236	22.280	0.278	0.572	0.292	0.137	33.130	0.100	1.715	0.122	0.120	0.354	0.041	0.041	0.180
CV (%)	0.700	0.400	0.800	0.300	0.800	8.200	3.900	31.500	0.200	19.600	4.500	8.200	10.900	3.700	6.700	13.100
*p*-value	<0.001	<0.001	<0.001	<0.001	<0.001	<0.001	<0.001	0.984	<0.001	0.045	<0.001	<0.001	<0.001	<0.001	<0.001	<0.001

Means in the same column followed by the same letter are not significantly different, while values with different superscript letters are significantly different according to Fisher’s test. **RSW**; root system width (cm), **RSD**; root system depth (cm), **CHA**; convex hull area (cm^2^), **TRL**; total root length (cm), **RBC**; root branch count, **TRV**; total root volume (cm^3^), **FSRL**; first seminal root length (cm), **SRA**; seminal root angle, **LA**; leaf area (cm^2^), **LN**; leaf number **RDM**; root dry mass (g), **SDM**; shoot dry mass (g), **RSM**; root–shoot mass ratio, **RMR**; root mass ratio (g g^−1^), **SMR**; shoot mass ratio (g g^−1^), **RTD**; root tissue density (g cm^−3^).

**Table 4 plants-11-02522-t004:** Mean values for root and leaf traits of nine citron watermelon accessions evaluated in rhizotrons under non-stressed conditions 35 days after planting.

	Below Ground								Above Ground		Allometry					
Accession	RSW	RSD	CHA	TRL	RBC	TRV	FSRL	SRA	LA	LN	RDM	SDM	RSM	RMR	SMR	RTD
WWM-09	9.875 ^c^	33.620 ^b^	1928 ^b^	70.990 ^b^	50 ^b^	1.753 ^bc^	2.458 ^bc^	65.470 ^a^	76.410 ^a^	11.000 ^ab^	4.489 ^b^	2.884 ^a^	1.553 ^ab^	0.608 ^ab^	0.392 ^ef^	2.584 ^a^
WWM-15	6.411 ^g^	25.860 ^f^	1020 ^f^	49.970 ^g^	39 ^f^	1.541 ^cd^	1.106 ^f^	60.600 ^a^	57.880 ^e^	7.000 ^e^	1.852 ^e^	2.042 ^bc^	0.905 ^g^	0.475 ^f^	0.525 ^a^	1.219 ^de^
WWM-37(2)	9.889 ^c^	31.260 ^d^	1594 ^d^	71.720 ^a^	36 ^i^	1.837 ^b^	2.611 ^b^	62.430 ^a^	71.820 ^b^	9.000 ^cd^	2.061 ^de^	1.800 ^c^	1.142 ^def^	0.533 ^de^	0.467 ^bc^	1.137 ^e^
WWM-39	8.448 ^e^	31.330 ^d^	1337 ^e^	66.480 ^d^	45 ^d^	1.596 ^bcd^	2.353 ^bc^	45.470 ^a^	55.240 ^g^	9.000 ^de^	2.735 ^c^	2.172 ^b^	1.266 ^cd^	0.558 ^cd^	0.442 ^cd^	1.717 ^cd^
WWM-41(A)	12.132 ^b^	32.350 ^c^	1702 ^c^	67.520 ^c^	46 ^c^	1.809 ^b^	1.724 ^d^	66.600 ^a^	63.470 ^c^	10.000 ^bc^	2.695 ^c^	2.291 ^b^	1.169 ^de^	0.538 ^de^	0.462 ^bc^	1.502 ^cde^
WWM-46	7.053 ^f^	23.040 ^g^	872 ^h^	48.060 ^i^	37 ^h^	1.265 ^e^	1.793 ^d^	53.030 ^a^	50.330 ^h^	10.000 ^bc^	2.508 ^cd^	1.844 ^c^	1.370 ^bc^	0.577 ^bc^	0.423 ^de^	1.992 ^bc^
WWM-64	8.958 ^d^	28.770 ^e^	812 ^h^	52.140 ^f^	38 ^g^	1.369 ^de^	1.884 ^d^	45.300 ^a^	56.730 ^f^	9.000 ^cd^	2.129 ^cde^	2.247 ^b^	0.950 ^fg^	0.486 ^f^	0.514 ^a^	1.555 ^cde^
WWM-68	8.644 ^de^	25.420 ^f^	920 ^g^	48.750 ^h^	41 ^e^	1.596 ^bcd^	1.520 ^e^	61.870 ^a^	39.870 ^i^	11.000 ^ab^	2.343 ^cde^	2.224 ^b^	1.050 ^efg^	0.512 ^ef^	0.488 ^ab^	1.466 ^cde^
WWM-76	12.879 ^a^	34.910 ^a^	2078 ^a^	65.660 ^e^	54 ^a^	2.193 ^a^	2.837 ^a^	56.970 ^a^	60.720 ^d^	12.000 ^a^	5.118 ^a^	3.007 ^a^	1.704 ^a^	0.630 ^a^	0.370 ^f^	2.355 ^ab^
l.s.d	0.367	0.496	16.730	0.161	0.467	0.228	0.155	33.710	0.483	1.715	0.560	0.288	0.185	0.037	0.037	0.481
CV (%)	2.300	1.000	0.700	0.200	0.600	8.000	4.500	34.200	0.500	10.300	11.300	7.400	8.700	4.000	4.800	16.300
*p*-value	<0.001	<0.001	<0.001	<0.001	<0.001	<0.001	<0.001	0.842	<0.001	<0.001	<0.001	<0.001	<0.001	<0.001	<0.001	<0.001

Means in the same column followed by the same letter were not significantly different, while values with different superscript letters are significantly different according to Fisher’s test. **RSW**; root system width (cm), **RSD**; root system depth (cm), **CHA**; convex hull area (cm^2^), **TRL**; total root length (cm), **RBC**; root branch count, **TRV**; total root volume (cm^3^), **FSRL**; first seminal root length (cm), **SRA**; seminal root angle, **LA**; leaf area (cm^2^), **LN**; leaf number **RDM**; root dry mass (g), **SDM**; shoot dry mass (g), **RSM**; root–shoot mass ratio, **RMR**; root mass ratio (g g^−1^), **SMR**; shoot mass ratio (g g^−1^), **RTD**; root tissue density (g cm^−3^).

**Table 5 plants-11-02522-t005:** Pearson correlation coefficients for evaluated traits (root system architectural traits, leaf traits, allometry) under non-stressed conditions (*bottom diagonal*) and water stress conditions (*top diagonal*).

Traits	RSW	RSD	CHA	TRL	RBC	TRV	FSRL	SRA	LA	LN	RDM	SDM	RSM	RMR	SMR	RTD
**RSW**	**1**	0.83 **	0.98 **	0.90 **	0.78 **	0.85 **	0.69 *	−0.02 ^ns^	0.82 **	−0.16 ^ns^	0.37 ^ns^	−0.39 ^ns^	0.82 **	0.83 **	−0.82 **	−0.49 ^ns^
**RSD**	0.83 **	**1**	0.93 **	0.99 **	0.97 **	0.86 **	0.87 **	−0.06 ^ns^	0.85 **	0.01 ^ns^	0.66 *	−0.22 ^ns^	0.93 **	0.94 **	−0.94 **	−0.31 ^ns^
**CHA**	0.80 **	0.90 **	**1**	0.97 **	0.89 **	0.89 **	0.79 **	−0.06 ^ns^	0.86 **	−0.08 ^ns^	0.51 *	−0.33 ^ns^	0.90 **	0.91 **	−0.90 **	−0.43 ^ns^
**TRL**	0.67 *	0.89 **	0.87 **	**1**	0.95 **	0.91 **	0.84 **	−0.06 ^ns^	0.91 **	−0.02 ^ns^	0.61 *	−0.25 ^ns^	0.92 **	0.93 **	−0.93 **	−0.39 ^ns^
**RBC**	0.70 *	0.75 **	0.79 **	0.53 *	**1**	0.77 **	0.82 **	−0.25 ^ns^	0.79 **	0.14 ^ns^	0.72 *	−0.15 ^ns^	0.93 **	0.93 **	−0.93 **	−0.21 ^ns^
**TRV**	0.83 **	0.82 **	0.89 **	0.70 *	0.73 *	**1**	0.79 **	0.23 ^ns^	0.92 **	−0.19 ^ns^	0.38 ^ns^	−0.36 ^ns^	0.78 **	0.80 **	−0.80 **	−0.62 *
**FSRL**	0.62 *	0.76 **	0.72 *	0.76 **	0.51 *	0.63 *	**1**	0.12 ^ns^	0.71*	0.20 ^ns^	0.73 *	−0.04 ^ns^	0.89 **	0.90 **	−0.90 **	−0.11 ^ns^
**SRA**	0.32 ^ns^	0.20 ^ns^	0.47 ^ns^	0.31 ^ns^	0.21 ^ns^	0.46 ^ns^	−0.09 ^ns^	**1**	−0.02 ^ns^	−0.58 *	−0.43 ^ns^	−0.50 *	−0.03 ^ns^	0.01 ^ns^	−0.02 ^ns^	−0.48 ^ns^
**LA**	0.41 ^ns^	0.70 *	0.71 *	0.80 **	0.30 ^ns^	0.46 ^ns^	0.54 *	0.38 ^ns^	**1**	−0.05 ^ns^	0.53 *	−0.10 ^ns^	0.70 *	0.74 *	−0.73 *	−0.44 ^ns^
**LN**	−0.07 ^ns^	0.22 ^ns^	0.06 ^ns^	0.28 ^ns^	0.23 ^ns^	−0.02 ^ns^	0.30 ^ns^	−0.54 *	−0.14 ^ns^	**1**	0.70 *	0.79 ^ns^	0.12 ^ns^	0.09 ^ns^	−0.09 ^ns^	0.80 **
**RDM**	0.64 *	0.68 *	0.79 **	0.49 ^ns^	0.91 **	0.68 *	0.67 *	0.18 ^ns^	0.38 ^ns^	0.07 ^ns^	**1**	0.53 *	0.65 *	0.66 *	−0.65 *	0.46 ^ns^
**SDM**	0.64 *	0.69 *	0.68 *	0.39 ^ns^	0.92 **	0.63 *	0.47 ^ns^	0.18 ^ns^	0.31 ^ns^	0.01 ^ns^	0.91 **	**1**	−0.28 ^ns^	−0.28 ^ns^	0.28 ^ns^	0.84 **
**RSM**	0.54 *	0.55 *	0.73 *	0.50 *	0.76 **	0.56 *	0.74 *	0.11 ^ns^	0.35 ^ns^	0.16 ^ns^	0.92 **	0.69 *	**1**	0.99 **	−0.99 **	−0.24 ^ns^
**RMR**	0.53 *	0.54 *	0.72 *	0.52 *	0.73 *	0.54 *	0.75 **	0.12 ^ns^	0.34 ^ns^	0.20 ^ns^	0.88 **	0.63 *	0.99 **	**1**	−1.00 **	−0.25 ^ns^
**SMR**	−0.53 *	−0.54 *	−0.72 *	−0.52 *	−0.73 *	−0.54 *	−0.75 **	−0.12 ^ns^	−0.34 ^ns^	−0.20 ^ns^	−0.88 **	−0.63 *	−0.99 **	0.23 ^ns^	**1**	0.25 ^ns^
**RTD**	0.34 ^ns^	0.41 ^ns^	0.51 *	0.26 ^ns^	0.73 *	0.27 ^ns^	0.52 *	−0.01 ^ns^	0.26 ^ns^	0.10 ^ns^	0.89 **	0.78 **	0.88 **	0.86 **	−0.86 **	1

**RSW**; root system width (cm), **RSD**; root system depth, **CHA**; convex hull area, **TRL**; total root length, **RBC**; root branch count, **TRV**; total root volume, **FSRL**; first seminal root length, **SRA**; seminal root angle **LA**; leaf area, **LN**; leaf number, **RDM**; root dry mass, **SDM**; shoot dry mass, **RSM**; root–shoot mass ratio, **RMR**; root mass ratio, **SMR**; shoot mass ratio, **RTD**; root tissue density, (***** and ****** denote significant at 5% and 1% probability levels, respectively; **ns**, non-significant).

**Table 6 plants-11-02522-t006:** Factor loadings, eigenvalue, Kaiser-Meyer-Olkin measure of sampling adequacy, percent and cumulative variation for root and leaf traits of nine citron watermelon accessions evaluated under non-stress and water-stress conditions.

	Non-Stress			Water-Stress		
Traits	PC 1	PC 2	KMO	PC 1	PC 2	KMO
RSW	0.792	0.283	0.725	0.898	−0.184	0.870
RSD	0.842	0.429	0.715	0.979	0.049	0.557
CHA	0.927	0.347	0.699	0.968	−0.082	0.575
TRL	0.735	0.570	0.621	0.991	−0.007	0.551
RBC	0.883	−0.121	0.757	0.946	0.184	0.658
TRV	0.802	0.384	0.720	0.906	−0.255	0.633
FSRL	0.798	0.084	0.667	0.888	0.182	0.753
SRA	0.278	0.438	0.499	−0.011	−0.682	0.243
LA	0.548	0.602	0.623	0.872	−0.040	0.530
LN	0.738	−0.416	0.621	0.008	0.936	0.316
RDM	0.942	−0.267	0.746	0.622	0.762	0.441
SDM	0.824	−0.187	0.633	−0.270	0.874	0.542
RSM	0.897	−0.351	0.630	0.955	0.085	0.676
RMR	0.885	−0.338	0.645	0.966	0.066	0.551
SMR	−0.885	0.338	0.645	−0.965	−0.063	0.556
RTD	0.761	−0.553	0.823	−0.369	0.882	0.453
Eigenvalue	10.233	2.369	−	10.319	3.656	−
Variability (%)	63.953	14.807	−	64.497	22.848	−
Cumulative (%)	63.953	78.760	−	64.497	87.345	−

**RSW**; root system width, **RSD**; root system depth, **CHA**; convex hull area, **TRL**; total root length, **RBC**; root branch count, **TRV**; total root volume, **FSRL**; first seminal root length, **SRA**; seminal root angle **LA**; leaf area, **LN**; leaf number **RDM**; root dry mass, **SDM**; shoot dry mass, **RSM**; root–shoot mass ratio, **RMR**; root mass ratio, **SMR**; shoot mass ratio, **RTD**; root tissue density.

**Table 7 plants-11-02522-t007:** Information on the source of seed, drought stress tolerance index and seed coat colour of citron watermelon accessions used in the study.

Accession	District	Village	Latitude and Longitude	STI	Seed Coat Colour
WWM-09	Capricorn	Moletjie-Moshate	23°36′55.9″ S 29°16′03.7″ E	0.452	nut brown
WWM-15	Capricorn	Turfloop	23°53′12.2″ S 29°44′52.2″ E	0.417	tomato red
WWM-37(2)	Capricorn	Ga-Molepo	24°01′11.1″ S 29°47′05.0″ E	0.392	purple violet
WWM-39	Capricorn	Ga-Mphela	23°43′19.2″ S 29°12′01.4″ E	0.431	ruby red
WWM-41(A)	Sekhukhune	Nebo	24°54′09.1″ S 29°46′15.8″ E	0.434	purple red
WWM-46	Sekhukhune	Nebo	24°54′07.2″ S 29°46′13.2″ E	0.459	signal red
WWM-64	Capricorn	Ga-Mphela	23°39′46.0″ S 29°19′16.4″ E	0.438	golden yellow
WWM-68	Capricorn	Ga-Manamela	23°43′01.7″ S 29°14′04.7″ E	0.468	brown-olive
WWM-76	Capricorn	Ga-Manamela	23°43′05.1″ S 29°14′01.3″ E	0.546	cream

**STI**; stress tolerance index.

**Table 8 plants-11-02522-t008:** Description of measured traits in citron watermelon accessions grown in a root rhizotron and assessed from 8 DAT to 35 DAT under water stress and non-stress conditions.

Trait(s)	Description	Unit(s)
Root system width (RSW)	Maximal horizontal distribution of a root system	cm
Root system depth (RSD)	Maximal vertical depth of a root system	cm
Convex hull area (CHA)	Area of the convex hull that encompasses the root system	cm^2^
Total root length (TRL)	Total sum of seminal and lateral root length	cm
Root branch count (RBC)	Number of lateral roots emerging from the primary root	-
Total root volume (TRV)	Total volume of the root system	cm^3^
Leaf area (LA)	Area of the leaf	cm^2^
Leaf number (LN)	Number of leaves	-
First seminal root length (FSRL)	Length of radicle (measured one day after germination)	cm
Seminal root angle (SRA)	Angle between the outermost left and right seminal roots	◦
Root dry mass (RDM)	Total dry mass of roots per plant	g
Shoot dry mass (SDM)	Total dry mass of shoots per plant	g
Root–shoot mass ratio (RSM)	Total root dry mass divided by shoot dry mass	-
Root mass ratio (RMR)	Dry mass of root divided by the total dry mass of entire plant	g g^−1^
Shoot mass ratio (SMR)	Dry mass of shoot divided by the total dry mass of entire plant	g g^−1^
Root tissue density (RTD)	Total root dry mass divided by root volume	g cm^−3^

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
