# Peer review of "Phenotypic Variability of Root System Architecture Traits for Drought Tolerance among Accessions of Citron Watermelon (Citrullus lanatus var. citroides (L.H. Bailey)"

_plants, 2022, doi:10.3390/plants11192522_

Round 1

Reviewer 1 Report

The MS is good study of the root system in depth with focusing different aspects of the root growth and potential for the drought stress of water melon. However, I do not agree with the conclusion made form the studied so many parameters. 

I would suggest that the conclusion both in abstract and in general shall be redrafted with focusing the parameters and its outcome for making decision for drought resistant which based on the root system observed characters.

Please add the citation Plant and Soil 267: 319-328 (2004).

Author Response

The response to reviewer 1 was uploaded as a PDF

Reviewer 2 Report

I believe this is valuable research. However, I think there are severe flaws in relation to data analysis. In general, the results are mainly expressed as one-way anovas, where a factorial design was conducted. Thus, the presentation of results as single effects (Figure 3, Figure 4) are incorrect. I suggest to thoroughly revising the data analysis and re-write the paper in those terms. Some suggestions are listed below: 

-        Paragraph in line 53, what constitutes a well-developed root system. What are the traits that explain better drought tolerance compared to the other crops.

-        In line 67 change “plant parts” to “plant organs”

-        Why only highly drought-tolerant accessions were used? Considering that it is important to understand root traits linked to drought tolerance or better performance during stress, it is highly informative to characterize root traits of less drought tolerant accessions.

-        In line 358, delete STI because it is described in table 7 caption.

-        Improve quality of figure 7.

-        In section 4.3, please explain how the gravimetric water content results were used for the selection of the mix used for this study.

-        In line 400, do you mean sowed? Because they are seeds not plants.

-        In lines 401 and 402: indicate the meaning of DAP

-        In line 402, irrigation was only done at sowing?

-        In section 4.3 please state that Gromor potting Mix corresponds to pine bark.

-        Figure 1, What the segmented vertical black lines represent? I don’t fully understand the exponential lines, please clarify in text.

-        Section 2.2, authors indicate that this is a factorial design, why results were not subjected to that analysis? For example, in table 1 I understand that comparisons were made between accessions of the same treatment. I think is more informative to know if the water stress treatment had an effect over the non-stressed plants and if this response depends on accessions. I advice to analyze data as factorial to assess if there is an interaction between accession and water treatment in root growth response.

-        In paragraphs between lines 127 and 136 there is mention regarding the effect of water stress on root traits, by comparing between NS and WS treatment, which is also displayed on figure 3. How this analysis was performed? Why the different accessions were not considered here? Do box plot shows mean values considering all accessions?. Considering that this should have been analyzed as a factorial design, it is highly probable that a significant interaction between water treatment and accession could have been observed in several root traits. This is why, combining all accessions for analysis is unreliable.

-        In Figure 3. In the y-axis change commas by dots.

-        Table 2. I suggest to add p-values for each trait and factor.

-        Section 2.3. For example, because there is a significant interaction between accession and water condition in the RSW parameter, the figure should explain this interaction. Thus, figure 4a is not accurate. This problem arises for the rest of the parameters were interaction is significant.

-        Section 2.4. There is no need to write in the text the correlation values, since they are already displayed in table 5. This makes the text more difficult to read.

-        In section 4.3 (lines 395-396) it is indicated that two experiments were performed. The first between September and October of 2021 and the second, between October and November of 2021. I don´t understand why there are two separate experiment and what is the difference between them. The results shown belong to the first or second experiment?

Author Response

Responses to reviewer's 2 comments were uploaded as a PDF

Reviewer 3 Report

The manuscript is well organized and the results are enriched. however, only 9 accessions were used and maybe the conclusions what they drew are not so accurate. 

Author Response

The response(s) have been uploaded as a PDF 

Reviewer 4 Report

The paper: Phenotypic variability of root system architecture traits for drought tolerance among accessions of citron watermelon (Citrullus lanatus var. citroides (L.H. Bailey), compares nine citron watermelon genotypes tolerant to water stress, in relation to arquiteture root traits under no-stress and water stress conditions.

In general, the paper is well written, but in the results, the authors need to check the indication of figures and tables. Also, the paragraphs are too long, and are dificult to read. In addition, some variables are presented without the correspondente units.

Below I included some observations and changes that need to be made:

Lines 13 to 15– In  Citron watermelon (Citrullus lanatus var. citroides) is an extremely drought-tolerant cucurbit crop widely grown in sub-Saharan Africa in arid and semi-arid environments characterized by drought”. - remove the word “an extremely” and “characterized by drought”.

Lines 17 and 18 – replace “in identifying” to  “to indentify”.

Line 44- “The fresh or dried vines are used as feed for domesticated animals” (add a reference here)

Line 45 – replace for: Also, the crop is efficent to absorb N

Line 47 – Fusarium (use italic)

Lines 62 to 63 – replace to “ Plant plasticity is importante to competitiveness...

Line 74 – (kg) – remove this

Line 83 – The objective of this work was....

Line 96 – Replace the word media for “substract” . This substract used is the same as sand pine bar, mix? if so, please add this information in the material and methods. For me, it is not clear.

Line 98 – how did you calculated the average growty rate? In the figures we can observed the growth rate and in table 1 only the average daily growth rate – Explain in the metodology, how did you calculated, not only in table one, but also in all other tables

Line 99 – (Figure 2a; Table 1) – in all this part, from lines 99 to 111, data are related to the figures and table 1.

Line 104 – replace (≤ 27.663 cm2 day-1) for (≤ 31.715 cm2 day-1)

Lines 106 t0 108 – “Accessions WWM-09, WWM- 41(A) and WWM-76 had average RBC ≈ 2 branches day-1, while WWM-15, WWM-37(2), WWM-39 and WWM-68 were forming approximately one branch per day (Figure 2i). – Check this information – it is confused.

Line 114 – replace (≤ 0.284 cm2 day-1) for (≤ 0.292 cm2 day-1) (Figure 2b, Table 1).

Line 139– remove the word “key”

Line 140 – increase the size of the letter in Figure 2.

Line 137 – Figure 1: increase the title of this figure.

From lines 152 to 183 – please, separe the paragraphs. It is very difficult to read and compare all data. Write several paragraphs for each variable. For example: For RDS.....

Another paragraph – The mean CHA... etc.

Also, include the unitis of each variable discussed. When the authors present the values of a variable, replace figure for table.

Line 157 – include access WWM-15 (This access was 6.411, so it was also below 8.644).

Line 159 – ....compared to WWM-15, WWM-46 and WWM-64, which recorded values ≤28.770 (cm) – include the accession WWM.68 ( this genotypes presented value of 25.420 cm, so it was lower than 28.770 cm.

Line 162 - ... For TRV, WWM-09, WWM-39, WWM-41(A), and WWM-68 recorded values ≥ 1.928 cm3 under WS... Include the accessions WWM-37(2) and WWM-76, as they also present values ≥ 1.928 cm3

Line 173 – “For RDM, accessions WWM-76 and WWM-09 recorded values ≥ 2.884 u der NS – I did not find this value in table 3

Line 175 – “ WWM-09, WWM-39, WWM-64 and WWM-76 recorded SDM values of = 2.172 g under NS condition”

(include the accessions WWM-41(A) and WWM-46)

Line 182 - . The mean root tissue density (RTD) – include between brackets the symbol RTD

Lines 232 and 233 – remove: “PC3 accounted for 9.76% of the total variation and was positively correlated with SRA, TRV and RBC.” – PC3 represents a very low variance. Also remove it from table 6 (lines 269 -270)

Line 235- remove RSM ( it is written twice), leave only one RSM

Line 235 – Leaf number (LN), RDM, ..... include (LN)

Line 275-276 – In figure 5. Indicate what is each variable. Ex. LA (Leaf área), etc.

Line 288 – remove”.”

Lines 297 to 298 – “On the contrary, according to our results, not all accessions evaluated in the presente study had increased root length under water stress” – this a mean of all accessions as presented in figure 4d – indicate table Table 4, not figure.

Lines 307 to 308 – Remove “Error! Reference source not found”

Line 309 – “watermelon exhibit higher above-and-below ground biomass under water deficit conditions as a drought-avoidance strategy”, but in this paper the root and above mass decreased under water stress (Check this afirmative).

Line 312 –“wake of increased weather conditions in the future” – please revise this frase. Wake conditions?

Lines 316-317 – “The shift in root growth and allometry observed in the present study can be explained by the “balanced growth” hypothesis wake of increased weather conditions in the future.” – but this dependo on the genotype

Lines 326 to 327 – “WWM-09, WWM-39 and WWM-41(A) had higher root: shoot ratios indicating their higher levels of drought tolerance.” – This information is from another pape ror frm your results? – please check.

“As evidenced by negative associations formed in PC biplots (Figure 5b) between SMR with TRL, CHA, RSD and RSW” – include the variable LA

Line 328 - Mandizvo, Odindo, Mashilo and Magwaza [26] – change to Mandizvoet al. [26]

Line 354 – made the same correction , change to Mandizvoet al. [26]

Line 358 – STI – include the formule to calculate (here or in Other part of the metodology.

Line 375 – figure 7 – in the squeme of thizotron – improve the quality of the squeme and in the text inside the squares.

Lines 279 to 384 – replace ‘medium’ to substract.

Line 407 – replace “terminated” to harvested.

Line 415 - [4:3] – remove brackets

Line 428 – “Error! Reference source not found” - correct

Author Response

Responses were uploaded as a PDF

Round 2

Reviewer 3 Report

The authora have addressed all my concerns. 

Author Response

Thank you for the constructive critical review of the initial manuscript draft. The issues raised in the first round of review have improved the quality of the manuscript.

Reviewer 4 Report

One phrase should be deleted between lines 432-433 "(Error! 432 Reference source not found". 

Also, the size of letters from figure 1 should be increased. 

Author Response

Responses to the comments were uploaded (see uploaded PDF)
